# Advances in Understanding the Links between Metabolism and Autophagy in Acute Myeloid Leukemia: From Biology to Therapeutic Targeting

**DOI:** 10.3390/cells12111553

**Published:** 2023-06-05

**Authors:** Ernestina Saulle, Isabella Spinello, Maria Teresa Quaranta, Catherine Labbaye

**Affiliations:** Istituto Superiore di Sanità, National Center for Drug Research and Evaluation, 00161 Rome, Italy; isabella.spinello@iss.it (I.S.); mariateresa.quaranta@iss.it (M.T.Q.)

**Keywords:** autophagy, hematopoiesis, acute myeloid leukemia, metabolism, therapy resistance

## Abstract

Autophagy is a highly conserved cellular degradation process that regulates cellular metabolism and homeostasis under normal and pathophysiological conditions. Autophagy and metabolism are linked in the hematopoietic system, playing a fundamental role in the self-renewal, survival, and differentiation of hematopoietic stem and progenitor cells, and in cell death, particularly affecting the cellular fate of the hematopoietic stem cell pool. In leukemia, autophagy sustains leukemic cell growth, contributes to survival of leukemic stem cells and chemotherapy resistance. The high frequency of disease relapse caused by relapse-initiating leukemic cells resistant to therapy occurs in acute myeloid leukemia (AML), and depends on the AML subtypes and treatments used. Targeting autophagy may represent a promising strategy to overcome therapeutic resistance in AML, for which prognosis remains poor. In this review, we illustrate the role of autophagy and the impact of its deregulation on the metabolism of normal and leukemic hematopoietic cells. We report updates on the contribution of autophagy to AML development and relapse, and the latest evidence indicating autophagy-related genes as potential prognostic predictors and drivers of AML. We review the recent advances in autophagy manipulation, combined with various anti-leukemia therapies, for an effective autophagy-targeted therapy for AML.

## 1. Introduction

Acute myeloid leukemia (AML), a common type of acute leukemia with poor survival and prognosis, is a clonal hematopoietic disorder affecting hematopoietic stem and progenitor cells, which results in the blockage of myeloid differentiation and the suppression of hematopoietic functions. AML, classified into subtypes according two major systems, the French–American–British (FAB) classification and the last World Health Organization (WHO) classification [1], are a group of aggressive hematologic malignancies resulting from acquired genetic mutations in hematopoietic stem cells that affect patients of all ages. Despite decades of research, standard chemotherapy still remains ineffective for some AML subtypes and/or is inappropriate for older patients or those with comorbidities. The molecular genetic characteristics of this disease, such as chromosomal translocations and recurrent mutations in the genes related to hematopoietic functions, have also directed decades of research in molecular targeted therapy for patients with AML [2]. However, poor clinical response and acquired resistance have also been observed in clinical applications of new agents/drugs mostly targeting the molecular aspects of leukemogenesis [3,4]. Furthermore, numerous studies have shown that disordered autophagy regulation is associated with cancer, including myeloid neoplasms [5,6], revealing the crucial role of autophagy in AML development and in the response to targeted therapies and indicating the autophagic pathways as potential targets, by using some inhibitors or activators, in AML treatment [7,8]. 

In this review, we report the current knowledge about the machinery and molecular mechanism of autophagy, its role during hematopoiesis and AML development and targeted therapy, and the recent advances in the therapeutic treatment related to the modulation of autophagy in leukemia. 

## 2. The Autophagy Process and Its Regulation

Autophagy is an evolutionarily conserved intracellular process essential to maintain cellular homeostasis and recycle cytoplasmic contents. As a process of digestion of intracellular contents, autophagy keeps the cytoplasm free of damaged organelles, aggregates and invading microbes, generating nutrients and energy to support cellular growth and survival. Furthermore, autophagy participates in cellular responses to external or internal stimuli due to hypoxia, genomic instability, metabolic stress, energy demand, and chemotherapy in cancer, including leukemia [9]. Autophagy is also a cell-death program, for which the molecular mechanisms remain to be further elucidated, as is also its interplay with apoptosis. Autophagy can act as either a pro-survival or pro-death process, both tissue and microenvironment specific [10].

### Molecular Mechanism for Autophagy

Autophagy is a catabolic pathway that initiates with the sequestration of cargoes (i.e., bulk cytoplasm, superfluous or damaged organelles, aberrant or dysfunctional proteins and protein aggregates, pathogens) into large double-membrane vesicles called autophagosomes that fuse with lysosomes to form the autolysosome where cargoes are degraded and recycled. Depending on the cargoes delivered to lysosomes for degradation, there are different forms of autophagy [11,12]. In mammalian cells, three major forms of autophagy have been described. Macroautophagy (also called autophagy) relies on cargoes degradation via autophagosomes to sequester and transport cargoes to the lysosome; microautophagy involves the direct uptake of cargoes through the invaginations of the lysosomal membrane; the chaperone-mediated autophagy (CMA) transports unfold proteins, one by one, directly across the lysosomal membrane through the interaction with chaperones (HSC70) that are recognized by the LAMP2A (lysosomal-associated membrane protein 2A) receptor located in the lysosome membrane [11,12,13]. In addition, a selective autophagy can recognize specific targets, such as damaged mitochondria (mitophagy), aggregated proteins (aggrephagy), and invading bacteria (xenophagy), to engulf, by isolation, the membrane and degrade the toxic materials within lysosomes [14]. Mitophagy, involved to maintain the number and integrity of the mitochondria engaged in active metabolism, is of particular importance for the lifelong maintenance of hematopoietic stem cells (HSCs) and leukemic stem cells (LSCs), whose resistance to therapeutic treatment is dependent on mitochondrial metabolism in AML [15]. 

Macroautophagy involves approximately 40 autophagy-related (*ATGs*) proteins, recruited in a pre-autophagosomal structure (PAS) responsible for autophagosome formation [16], which regulate each step of the autophagic, also called “canonical”, process [11,12]. The “non-canonical” process is when none of the ATGs activities are required, such as in chaperone-mediated autophagy and microautophagy [12,13]. Overall, *ATGs* proteins are promising biomarkers and therapeutic targets in different tumors [17], and their expression is epigenetically controlled by microRNAs (miRNAs) and long non-coding RNAs (lncRNAs) [18,19]. The autophagic process consists of several steps and is mainly dependent on six protein complexes: the Unc-51-like autophagy activating kinase 1 (ULK1) complex, central in the initiation stage of autophagy, class IIIPI3K (phosphoinositide 3-kinase) complex, ATG9A complex, ATG2-ATG18/WIPI complex, and finally the two ATG8/LC3 and ATG12-ATG5-ATG16L1 conjugation complexes. The process of autophagy includes five steps: initiation, phagophore nucleation, elongation and autophagosome formation, autophagosome–lysosome fusion, and cargo degradation, where autophagy-related genes (*ATGs*) play a key role in the entire pathway. Starting with the induction and nucleation of the membrane to form an autophagosome that, by sequestering portions of the cytoplasm, engulfs unfolded protein aggregates and organelles [20,21], ULK1 complex, via its kinase activity, integrates nutrient sensing pathways for growth factors, amino acids, and energy by regulating the activity of this autophagy-inducing complex. ULK1 phosphorylation is under the control of two mediators of autophagy, AMP-activated protein kinase (AMPK), a major-energy sensing kinase [22,23], and mTOR (mammalian target of rapamycin), a negative regulator of autophagy, as described in [24]. 

In the presence of nutrient abundance or growth factor signaling, mTOR complex 1 (mTORC1) inhibits the activity of the ULK regulatory complex through the phosphorylation of ULK1, preventing autophagy. Conversely, altered intracellular levels of ATP are detected by AMPK which, in turn, activates the ULK1 complex, including ULK1/ATG1, ATG13, FIP200/ATG17 (protein that interacts with the FAK family kinase), and ATG101, resulting in the activation of the autophagic process [22,23]. Under conditions of nutrient deprivation, AMPK acts as a negative regulator of mTORC1 by activating mTORC1 inhibitors, and thereby promotes mTOR deactivation and ULK1 phosphorylation leading to the activation of the ULK1 complex driving autophagosome formation [21,22]. The supramolecular assembly of multiple ULK kinase complexes generates a scaffold for the formation of a phagophore, the precursor of an autophagosome [21,22]. The autophagy-specific phosphatidylinositol 3-kinase complex (PI3K-III) also participates in the phagosome nucleation by catalyzing the synthesis of phosphatidylinositol 3-phosphate on nascent autophagosomal membranes. The PI3K-III complex, including Beclin1/ATG6, ATG14, VPS34, VPS15 (vacuolar protein sorting 34 and 15), and AMBRA 1 (Activating molecule in beclin1-regulated autophagy), once activated by the ULK1 complex, phosphorylates the phosphatidylinositol diphosphate (PIP2) in triphosphate PIP3, an increased concentration of which causes the recruitment of proteins, such as DFCP1 (double FYVE domain-containing protein1), WIPI/ATG18 (WD-repeat domain phosphoinositide interacting protein1), ATG2, and ATG9A positive vesicle, at the site of phagophore formation [25]. 

Autophagy can be inhibited by the binding of the Bcl2 protein, involved in cell death regulation by controlling the mitochondrial membrane permeability, to Beclin1 [26], a protein that interacts with AMBRA 1 to promote autophagosome formation via the regulation of the activity of a Beclin1/Vps34 complex [12,22]. 

The subsequent step of elongation for autophagosomal formation is mediated in the membrane by two ubiquitin-like conjugation complexes ATG12-ATG5-ATG16L1 and members of the microtubule-associated protein 1 light chain 3 (LC3) family (homologs of yeast Atg8). The association of the ATG12-ATG5 dimeric complex with the membrane, via the attachment of ATG16L1, is necessary for its functional activity as an E3 enzyme for LC3 conjugation to phosphatidylethanolamine (PE) on the autophagosomal membrane. The ATG8/LC3 conjugation to PE in membrane complexes is involved in the conversion of LC3-I to the lipid form LC3-II by the proteolytic cleavage of ATG4 (a cysteine protease), for the subsequent conjugation to the membrane that is mediated by the ATPase ATG7 (E1-like enzyme) and ATG3 (the E2-like enzyme) [27]. LC3 plays a key role in autophagy, participating in multiple steps of autophagosome biogenesis, and it is critical in promoting autophagosome–lysosome fusion [26,27,28]. In addition to LC3, the ATG12 conjugation complex, with ATG7 (E1-like protein) and ATG3 (E2-like protein), helps in the recruitment and conversion of LC3-I to LC3-II [27,28,29].

In selective autophagy, during the nucleation and elongation steps, some cargo receptors are involved in LC3-dependent cargo selection [29,30]. The engagement of ubiquitin-dependent autophagy receptors is dictated by the ubiquitylation of the cargo destined for degradation. Different cargo receptors, such as p62/sequestosome-1 (SQSTM1/p62), neighbor of BRCA1 (NBR1), Optin (OPTN), TAX1BP1, CALCOCO2/NDP52, and TOLLIP [30], are involved in the transport of cargoes to the nucleation site by their LC3-interacting region (LIRs) or ATG8-interacting motifs (AIMs) which facilitate cargo selection as well as selective autophagy [31]. Once protein recruitment and autophagosome formation are completed, the mature autophagosome with its contents fuses with lysosomes to become an autolysosome. Then, the autophagic cargo is degraded in the acidic environment of the lysosome by proteases cathepsins B and L, and the degraded products are transported into the cytoplasm to be reused in biosynthetic processes or to generate energy [32]. 

Autophagy is not only a process of digestion of intracellular contents that generates nutrients and energy to support cellular growth; autophagy is also involved in important cellular responses to external or internal stimuli and is dependent on the microenvironment [33]. In this context, altered gene expression and genetic mutations of several members pertaining to the protein complexes involved in the initiation phase of autophagy, the multiple steps of autophagosome biogenesis, and the autophagosome–lysosome fusion, as found in AML [5,6,7,8], can produce significant molecular and cellular anomalies and downstream effects along the all autophagy process involved in cancer formation and progression [12,17,34]. 

## 3. Autophagy in Hematopoiesis

### 3.1. Hematopoietic Stem Cells (HSCs)

Hematopoiesis is a finely controlled process, based on the regulation of quiescence, self-renewal, and the multilineage differentiation of hematopoietic stem cells (HSCs). HSCs reside in specialized niches in the bone marrow (BM) in a metabolically inactive quiescent state. In response to specific stimuli, HSCs show the peculiarity of self-renewal and/or differentiate into hematopoietic progenitors cells (HPCs), which, in turn, give rise to mature hematopoietic cell lines. Metabolism and autophagy play a crucial role in maintaining hematopoietic system homeostasis, providing self-renewal and multipotent HSC differentiation potential. Alterations in both the metabolic state and the regulation of autophagic machinery are implicated in the development of hematological neoplasms, especially in leukemogenesis [5,6,7,8]. Healthy HSCs rely primarily on bone marrow oxygen tension to maintain their stemness, by altering their gene expression to fit a metabolic profile based predominantly on anaerobic glycolysis [35,36]. Metabolomics analysis indicates that HSCs, compared to more differentiated progenitors, increase glycolysis and repress the influx of glycolytic metabolites into the mitochondria, through the regulation of pyruvate dehydrogenase kinase (PDK) activity by HIF-1α [37]. In addition, the production of glycolytic adenosine triphosphate (ATP) by the HIF-1α/PDK regulatory system is necessary to maintain HSC quiescence. Several studies indicate autophagy as a support of the glycolytic flow protecting HSCs from the metabolic stress and supporting their expansion into the bone marrow, thus reducing the possibility of HSC depletion [38]. The differentiation of HSCs is then associated with a metabolic shift from anaerobic glycolysis to mitochondrial-based energy via the tricarboxylic acid (TCA) cycle for ATP production and oxidative phosphorylation (OXPHOS) [35,36,37]. Additionally, a higher level of autophagic activity has been observed in HSCs and is associated with decreased HSC differentiation into progenitors [38,39]. Quiescent HSCs contain a high number of mitochondria, comparable to that of HPCs, involved in the maintenance of HSCs [40]. Mitochondria participate in antioxidant processes to reduce ROS levels. ROS modulate the activity of protein kinases (c-Kit, Akt, and MAPK) and transcription factors (p53, HIFs, and FOXOs), crucial for HSCs maintenance [41,42]. Furthermore, the inhibition of the Akt-mTOR pathway induces autophagy and suppresses OXPHOS, thus contributing to maintaining HSC quiescence. Under conditions of reduced cellular energy or low nutrients, HSCs can activate AMPK, thereby resulting in the inactivation of mTOR, and rapidly driving the induction of autophagy [43,44].

The targeted deletion of liver kinase B1 (*LKB1*) tumor suppressor, one of the AMPK upstream kinases, is critical for the maintenance of energy homeostasis in HSCs by a direct activation of AMPK, determining an inhibition in mTOR signaling [45]. Furthermore, *LKB1* loss in HSCs causes dysfunctional mitochondria, leading to pancytopenia due to reduced levels of ATP, fatty acids, and nucleotides [46]. A study in a mouse model, conditionally deleted for *LKB1*, suggests that the inactivation of mTOR in mice could be obtained through both AMPK-dependent and independent pathways, leading to autophagy activation to compensate for the acute loss of *LKB1* [46]. *LKB1* also participates in mitochondrial biogenesis, as *LKB1*-knockout HSCs result in a severely compromised HSC phenotype with defects in mitochondrial number and function [46]. 

Several autophagy proteins have been identified that play an important role in regulating stem cell quiescence, such as FOXO3a, which regulates autophagy levels in HSCs under metabolic stress [41]. In particular, mice deficient in *FOXO3a* showed a high reduction in autophagy ability to protect HSCs. FIP200, a component of the ULK1 initiation complex, has also been associated with HSC maintenance [47]. In mice, *FIP200* depletion has resulted in an increased proliferation of HSCs, in which mitochondrial mass was higher than in HSCs without depletion [47]. During normal hematopoiesis, mitophagy satisfies the metabolic needs of hematopoietic cells, from HSCs to mature blood cells, and, through the regulation of ROS, controls the degree of DNA damage (Figure 1). In particular, mitochondrial content and activity, involved in the maintenance and differentiation of hematopoietic stem/progenitor cells (HSPCs), play a key role in cellular metabolism and redox balance. Maintaining basal autophagy levels is essential for HSCs to eliminate damaged mitochondria and ROS accumulation. The normal modulation of mitophagy level is responsible for the selective clearance of mitochondria, and the Pten-induced putative kinase 1 (PINK1) and the ubiquitin-protein ligase Parkin have critical roles in this process [48,49]. During mitophagy, the degradation of Pink1 is impaired, leading to the accumulation and activation of this kinase on the mitochondrial outer membrane. PINK1 interacting with the external mitochondrial membrane and with Parkin, and ubiquitin ligase E3, drives mitochondria to autophagosomal degradation [49]. In the hematopoietic system, AAA+-ATPase (ATAD3A), an important regulator of mitophagy, facilitates the import of PINK1 into the mitochondria [50]. The deletion of *ATAD3A* results in the accumulation of PINK1 and the excessive activation of mitophagy in HSCs, resulting in mitochondrial depletion, the blockage of differentiation, and enlargement of HSC populations. In *ATAD3A* -deficient mice, the deletion of *Pink1* was able to correct mitophagic defects and then restore the pool of HSCs and their differentiation [50]. In addition, the genetic deletion of *ATG7* or other autophagy-related genes, such as *ATG12* or *ATG5*, is associated with increased mitochondrial content. In detail, the deletion of *ATG5* in hematopoietic cells causes a defect in the clearance of damaged mitochondria and a reduced HSC reconstitution potential [51]. The conditional knockout of *ATG12* impaired HSC reconstitution potential, which resulted in premature HSC aging [52]. *ATG12* -deficient HSCs exhibit an increase in active mitochondria with high ΔΨmt, indicating that ATG12-mediated autophagy is involved in their clearance. Similarly, the deletion of *ATG7* resulted in the crowding of mitochondria and ROS, as well as increased proliferation and DNA damage, resulting in the loss of HSC maintenance, hematopoietic abnormalities, and the failure of HSC engraftment after transplantation. Consistent with loss of HSC functions, the production of both lymphoid and myeloid progenitors was impaired in the absence of ATG7 [53]. A recent study reported that the deletion in mice of a membrane-associated E3 ubiquitin ligase subunit, Macrophage Erythroblast Attacher (*MAEA*), highly expressed in HSCs, severely impairs HSC quiescence and leads to lethal myeloproliferative syndrome. The deletion of *MAEA* destabilizes the surface expression of hematopoietic cytokine receptors and impairs autophagy flux in HSC, but not in their mature progeny. This protective mechanism affects HSC quiescence and function by limiting cytokine receptor signaling and autophagy [54]. The high number of lysosomes and their activity are critical in order to complete mitochondrial clearance in HSCs and to preserve their quiescent characteristics [55,56]. 

### 3.2. Erythrocytes

Autophagy plays an essential role during erythrocyte maturation. Reticulocytes, with cellular functions of hemoglobin production and the absorption of iron, use autophagy to remove mitochondria and other organelles such as ribosomes [54] before their terminal maturation. Once in the peripheral blood, the reticulocytes transport oxygen from the lungs to peripheral tissues to assist in metabolic processes. After “cleaning” a cell to remove superfluous cytoplasmic content, reticulocytes use the macromolecules produced during the catalysis to sustain the energy demand required for their morphological changes. In this context, several *ATG* -associated genes, such as *ULK1 (ATG1)*, *ATG7*, and *ATG5*, are critical for the clearance of mitochondria and ribosomes [57,58,59]. During reticulocyte maturation in mice, BNIP3L (BNIP3-like protein, also known as NIX), another HIF-1-induced target, is also required for programmed mitochondrial clearance by autophagy [60,61]. BNIP3 competes with Beclin 1 for binding Bcl-2 and thus releases Beclin 1 for participating in mitophagy. In addition, *ULK1*-deficient mice have impaired mitochondrial clearance during erythrocyte maturation phases [62]. GATA-1, the master regulator transcription factor of erythropoiesis, activates the fundamental autophagy component microtubule-associated protein 1A/1B-light chain 3 (LC3), and *ATG* related genes, *ATG4* and *ATG12*, as well as lysosomal genes in primary human erythroblasts [63]. More recently, Stolla et al. have shown that ATG4A is a cell type-specific regulator of autophagy in erythroid development [64]. ATG4A expression is necessary for the temporal induction of autophagic flux and mitochondrial clearance in terminally differentiated orthochromatic erythroblasts [64]. 

### 3.3. Megakaryocytes and Platelets

Megakaryocytes, indispensable for hemostasis, wound healing, angiogenesis, inflammation, and innate immunity, are the precursors of blood platelets required for the formation of a thrombus or blood clot. Megakaryocytic maturation, as for erythroid maturation, is regulated by GATA1, which induces the expression of several autophagic genes [65]. Reduced or abolished autophagy alters megakaryopoiesis and platelet formation [66,67,68]. Cao et al. demonstrated that autophagy deficiency in an *ATG7* hematopoietic conditional knockout mouse model causes mitochondrial and cell cycle dysfunction, resulting in defective megakaryopoiesis, megakaryocyte differentiation, and thrombopoiesis inhibition, leading to abnormal platelet production and functions [67]. The deletion of *ATG7* in mature megakaryocytes and platelets results in deregulated hemostasis, while platelet number and size are not changed [69]. Further investigation demonstrated abnormal aggregation and cargo granule packing in these platelets [69]. Therefore, it appears that autophagy is most likely necessary for the initial phase of megakaryocyte development and for normal platelet function. The disruption of the autophagic flux mediated by pharmacological inhibition leads to defects in platelet aggregation and adhesion, as shown by the bleeding for an extended time and platelet aggregation found in Becn1 heterozygous knockout mice [70]. Zhang et al. showed that hypoxia activates mitophagy in platelets in a FUNDC1-dependent manner, regulating mitochondrial quality and quantity, and controlling platelet activation during ischemia/reperfusion injury [71]. BNIP3L (also known as NIX), a previously characterized mitophagy receptor involved in erythrocyte maturation, also mediates mitophagy in platelets. The genetic ablation of *Nix* impairs mitochondrial clearance, platelet activation, and thrombosis. The loss of *Nix*, in vivo, by inhibiting autophagic degradation of the mitochondrial protein Bcl-xL increases the life span of platelets [71]. 

### 3.4. Granulocytes

Recent transcriptomic analyses have identified the expression patterns of autophagy genes during the monocytic (Mo) and granulocytic (G) differentiation of CD34+ hematopoietic stem and progenitor cells (HSPCs) and have identified, in particular, 22 autophagy genes with significant divergent roles during the Mo and G differentiation of HSPCs, which may reveal a specific autophagy-related gene signature leading to monocytes and granulocytes [72]. Granulocytes, including neutrophils, basophils, and eosinophils, are involved in the innate immune system to respond to infection or allergens, acting through phagocytosis and the release of enzymes to kill or neutralize microorganisms and toxins. The deletion of autophagy genes *ATG5* and *ATG7* has revealed their function in the regulation of neutrophil precursors, and their maturation through lipophagy, a specific type of autophagy that removes lipid droplets to generate free fatty acids. *ATG7* -deletion blocking lipophagy reduces oxidative metabolism and increases glycolysis, leading to the accumulation of lipid droplets and immature neutrophils [73]. Furthermore, the deletion of *ATG5* and *ATG7* results in reduced granulocytic release, impairing neutrophil function during inflammation [73]. In a recent study, ATG5 has been involved in the regulation of eosinophil differentiation, both in mouse and human models. Eosinophils in *ATG5* -knockout mice and *ATG5* -low-expressing human eosinophils have an enhanced degranulation capacity, suggesting that the effector functions of eosinophils are influenced by autophagy [74]. Like neutrophils, eosinophils can activate a form of necroptosis, a regulated necrosis mediated by death receptors [75], to stimulate an inflammatory response under specific conditions. This form of eosinophil cytolysis involving the formation of extracellular DNA traps, also known as ETosis [76], is mediated by the generation of ROS produced by NADPH-oxidase [77]. Adhesion-induced eosinophil cytolysis takes place through RIPK3-MLKL-dependent necroptosis, which can be counter-regulated by autophagy. Pretreating eosinophils with rapamycin, an inhibitor of mTOR, resulted in increased autophagic activity together with a significant reduction in vacuolization and cell death [77,78]. 

### 3.5. Monocytes

Autophagy is essential during monocytic differentiation and for the functions exerted by monocytes/macrophages in tissue homeostasis and protective immunity [79]. When tissue damage or infection occurs, resident or bone marrow monocytes are rapidly enrolled in the tissue where they mature into macrophages or dendritic cells by inducing autophagy. In absence of a maturation signal, monocytes inhibit autophagy and undergo apoptotic cell death. Beclin1 and Bcl2 proteins are involved in macrophage autophagy activation, while ATG5 calpain-cleaved is involved in autophagy inhibition [80]. The induction of xenophagy, a specific type of autophagy that removes invading organisms, is involved in removing pathogens. Xenophagy can be triggered through TLR4-mediated signaling [81], or by nucleotide-binding oligomerization domain-like (NOD) receptors, recruiting ATG16L1 at the bacteria entry site on the macrophage membrane [82]. Emerging evidence has shown that macrophage autophagy is important in macrophage polarization, chronic inflammation, and organ fibrosis [83]. 

## 4. Autophagy in AML

Studies have shown the role of autophagy in the life-long maintenance of HSCs [6,7,35,36], cancer progression, and chemoresistance [12,33,34,35]. The cytogenetic and genomic alterations/molecular abnormalities that accompany the malignant transformation of HSCs or HPCs in leukemic stem cells or leukemic cells have allowed the development of therapies for AML [3,4,5]. However, even though most leukemic cells are eliminated during treatment, a small population of persistent AML stem cells (LSCs) and therapy-resistant cells are not fully eradicated with current treatments [3,4], leading to treatment failure and relapse in AML patients. Autophagy plays different roles in AML, promoting leukemic cell survival or exerting an antileukemic effect [84,85,86]. Autophagy is essential for the initiation and maintenance of LSCs and for inducing chemoresistance during AML treatment, determining cell survival in these cells [84,85,86]. Therefore, targeting autophagic pathways in AML, through their selective inhibition, may help to overcome the therapeutic resistance and survival of LSCs involved in relapse [85,86,87,88]. In opposition to its pro-survival role, autophagy is also involved in the degradation of the fusion proteins PML-RARA and BCR-ABL which contribute to antileukemic responses in acute promyelocytic leukemia (APL) and chronic myeloid leukemia (CML), respectively [86,89]. In this context, autophagy inducers may be useful in APL and CML therapies [89]. Importantly, the induction of autophagy-mediated cell death can bypass the apoptotic death program, often inefficient in cancer cells, to eliminate/kill the refractory cancer cells [10,86,90]. Then, different approaches have been proposed to modulate autophagy, through its inhibition or activation, to overcome AML chemoresistance [8,86,90], as a promising target for antileukemic therapy.

### 4.1. Autophagy and AML Stem Cells

During leukemic development, LSCs can adapt their metabolism and autophagy mechanisms to supply high energy and nutrients required for LSCs proliferation and survival under conditions of nutrient deficiency, starvation, hypoxia, or during chemotherapy treatments [7,91,92,93,94].

Mitophagy, mitochondrial function, and integrity may affect the viability, proliferation, and differentiation potential and longevity of normal hematopoietic stem cells (HSCs) [95], and are important in the survival strategy of AML stem cells (LSCs).

In particular, the adenosine5′-monophosphate (AMP)-activated protein kinase (AMPK), a protein complex fundamental in mitochondrial metabolism and mitophagy, is constitutively activated in LSCs, increasing mitochondrial clearance to support LSCs growth and survival through its downstream target FIS1, the mitochondrial fission 1 protein component of a mitochondrial complex that promotes mitochondrial fission [96]. AMPK/FIS1-mediated mitophagy is required for the self-renewal and survival of LSCs [96]. An overexpression of FIS1 was also found in AML cells while FIS1 depletion impairs mitophagy, weakening the self-renewal capacity of LSCs and determining the induction of myeloid differentiation by GSK3 inactivation (glycogen synthase kinase 3), thus indicating mitophagy as being a regulatory mechanism for the progression of AML [96]. More recently, the loss of sequestosome 1 (*SQSTM1*), also known as p62, a selective autophagy receptor crucial for the development and progression of AML in vivo, induces the accumulation of damaged mitochondria and mitochondrial superoxide, thus compromising the survival of leukemia cells. Then, the loss of *SQSTM1* impairs leukemia progression in AML mouse models, underlying the role of mitophagy in the survival of LSCs [97]. Altogether, these studies demonstrate that enhanced autophagic activity of LSCs is required for malignant progression into AML.

However, in contrast to the autophagy activation observed in AML, a loss of autophagy in healthy HSCs triggers the expansion of a population of progenitor cells in the bone marrow, giving rise to severe and invasive myeloproliferation, such as in human AML [53]. This apparent paradox can be explained by the distinct roles that autophagy can play during AML progression, which may be different at various stages of leukemogenesis [98,99]. Autophagy in normal HSCs may prevent the onset of cancer, as a tumor suppressive mechanism. Indeed, autophagy removes damaged organelles, such as mitochondria, and protects hematopoietic cells from genomic instability and inflammation, thus preventing the onset of leukemia. Particularly, increased DNA damage, high ROS levels, aneuploidy, and an aberrant accumulation of p62/SQSTM1 have been correlated with an impaired autophagic process, indicating a key role of autophagy in preventing tumor initiation [100]. Conversely, in established cancer, autophagy may function as a favorable pathway that promotes survival and tumor growth, by helping tumor cells to escape metabolic stress and death stimuli.

Some studies have also shown that the activation of autophagic flow plays a role only in the initiation of AML, with a transformation from HSCs to LSCs, and, therefore, after this phase, autophagy is not required for disease maintenance [101]. Studies in MLL-AF9 AML, the most common alteration in childhood AML, indicate ATG5 or ATG7 as necessary for the onset of AML, but once the leukemic condition is established, autophagy is not required for LSC function in vivo [101,102]. However, in a different MLL-ENL AML mouse model, Atg5 or Atg7 knockout reduced the number of functional LSCs, increased mitochondrial activation and ROS levels in these cells, and prolonged the survival of leukemic mice [103]. In this context, during the leukemogenesis process, histone methylation can regulate core autophagy effectors and upstream autophagic regulators such as ATG 5 and ATG7 to influence autophagy level indirectly [104]. Together, these studies suggest a highly complex and context-dependent role for autophagy in leukemic transformation with respect to the maintenance properties of LSCs in AML. 

The dual role of autophagy in AML, as a promoter or suppressor of cancer in AML, is still a matter of debate. Studies have shown that autophagy can act as a pro or anti-proliferative mechanism depending on the lineage and the molecular genotypic context of the disease, reflecting the degree of heterogeneity of AML [105].

### 4.2. Regulation of Autophagy Genes in AML Cells

Numerous studies have shown that increased autophagy in AML cells confers protection from chemotherapeutic treatment and promotes AML cell survival. 

Increased ATG7-mediated autophagy has been associated with poor clinical outcomes and a short duration of remission in AML patients [106]. More recently, some proteins involved in leukemic cell survival, and overexpressed in AML, have been related to ATG overexpression, underlying the interplay between autophagy and protein overexpression promoting leukemic cell survival [8]. Hu et al. have shown that a high expression of SIRT1 (Sirtuin 1), a key player in mitochondrial biogenesis and autophagy-related protein, is associated with high CXCR4 expression, a negative prognostic marker in AML, and with other autophagy-related proteins such as ATG5 and LC3 in primary human AML samples, indicating a potential role of the SDF-1α-CXCR4 signaling pathway in autophagy induction in AML cells, which further promotes their survival under stress [107]. 

The transient receptor potential melastatin 2 (TRPM2) ion channel, involved in maintaining cell survival following oxidant injury, is highly expressed in AML [108]. By performing *TRPM2*-depletion, Chen SJ et al. [108] have shown that ULK1, Atg7, and Atg5 protein levels are decreased in *TRPM2*-depleted cells, leading to autophagy inhibition. Importantly, the depletion of *TRPM2* in AML inhibits leukemia proliferation and increases the doxorubicin sensitivity of AML cells [108]. 

Functional studies in normal CD34+ CB cells indicated that the inhibition of VMP1 expression reduced autophagic flux, with decreased hematopoietic stem and progenitor cell (HSPC) expansion, delayed differentiation, increased apoptosis, and impaired cell function and in vivo engraftment. Similar results were observed in leukemic cell lines and primary AML CD34+ cells. Furthermore, ultrastructural analysis indicated that leukemic cells overexpressing VMP1 have a reduced number of mitochondrial structures, and the number of lysosomal-degrading structures has increased. VMP1 (vacuole membrane protein-1) overexpression increased autophagic flux and improved mitochondrial quality, which coincided with an increased threshold for venetoclax-induced loss of mitochondrial outer membrane permeabilization (MOMP) and apoptosis in leukemia cells [109]. 

Heterozygous deletions, missense mutations, or changes in the number of copies of key autophagy genes have been found with a high frequency in AML patients, especially AML patients with complex karyotypes [5,103]. In particular, a heterozygous chromosomal loss of 5q, 16q, or 17p correlates with regions encoding autophagy genes *ATG10* and *ATG12*, *GABARAPL2* and *MAP1LC3B*, or *GABARAP*, respectively [103], and several others autophagy genes have a low level of expression in human AML blasts, a decreased autophagic flow, and high levels of ROS [103]. In addition, a study suggested that key autophagy genes such as *ULK1*, *ATG3*, *ATG4D*, and *ATG5* were significantly downregulated in primary AML cells compared to normal granulocytes [110].

### 4.3. Autophagic Biomarkers

Significant progress has recently been made to identify specific autophagy-related genes for the prediction of clinical outcomes in AML. Along with the *ATG* genes previously described, several microRNAs implicated in leukemogenesis and chemoresistance have been also involved in the activation of autophagy, and may be used as biomarkers [111]. In particular, miR-17-5p overexpression in leukemia promotes AML proliferation by inhibiting autophagy through BECN1 targeting [112,113,114]. Ganesan et al. demonstrated that stromal cells downregulate miR-23a-5p levels in leukemic cells, leading to the upregulation of protective autophagy in these cells, thereby increasing their resistance to chemotherapy toxicity [115]. MiR-143 overexpression was shown to enhance the sensitivity of AML cells to the cytotoxicity of cytarabine (Ara-C) treatment by inhibiting autophagy through ATG7 and ATG2B targeting [116]. An overexpression of miR-15a-5p is involved in the chemoresistance of AML patients, through autophagy-related genes *ATG9A*, *ATG14*, *GABARAPL1*, and *SMPD1* targeting AML cells [117].

Recent advances in bioinformatics have yielded an autophagy-related signature that can help to predict overall survival (OS) and/or the clinical outcomes of AML patients. Several studies have shown that the progression of AML depends on the autophagy-associated gene signature [118]. A recent bioinformatics study has built a model containing 10 autophagy-related genes to predict the survival of AML patients, showing that groups at high risk of AML have an increased expression of immune checkpoint genes and a higher percentage of CD4 T and NK cells [119]. In addition, this study was able to predict OS in AML through the signature of 10 genes, indicating this model as an effective prognostic predictor for AML patients, useful to guide patient stratification for immunotherapies and drugs [119]. The bioinformatics study LASSO Cox regression that identified a critical risk signature for AML, consisting of the autophagy genes *BAG3*, *CALCOCO2*, *CAMKK2*, *CANX*, *DAPK1*, *P4HB*, *TSC2*, and *ULK1*, had excellent predictive power for AML prognosis [120]. Notably, the immunosuppressive cytokines were found to be significantly increased in the tumor microenvironment of patients with a high-risk of AML, predicted on the basis of the autophagy-related signature of these patients [120]. However, the prognostic value of the ATG signature in the clinical setting is still debated. Therefore, the roles of the ATG signature and autophagy in the pathogenesis of AML should be further investigated.

In addition, an interesting study indicated that an autophagy-related lncRNA signature containing six lncRNAs (*HYMAI*, *MIR155HG*, *MGC12916*, *DIRC3*, *C1orf220*, and *HCP5*) may have an important prognostic value [121]. A recent study indicated four autophagy-associated lncRNAs (*MIR133A1HG*, *AL359715.1*, *MIRLET7BHG*, and *AL356752.1*) as a signature to potentially use as a biomarker to predict the survival of AML patients [122]. 

Altogether, these data indicate that the role of autophagy in tumor development clearly depends on the type of AML and the stage of tumor development. Furthermore, autophagy may provide cancer cells with a survival strategy, suggesting a therapeutic use for autophagy inhibition. On the other hand, autophagy can induce cell death, pointing to autophagy activation as a novel strategy in cancer therapy. Therefore, it is necessary to determine the role played by autophagy in the molecular subtypes of AML, or the degree of tumor development, to verify if its modulation could lead to benefit for the treated patient.

### 4.4. Autophagy and Genetic Alterations in AML

The AML phenotype results from multiple molecular, genetic, and epigenetic alterations affecting the differentiation, proliferation, and apoptosis of myeloid progenitors. The World Health Organization has classified AMLs according to the presence of particular genetic alterations, frequently originating from chromosomal translocations or other genome rearrangements such as t(8;21), t(15;17), inv (16), inv(3), t(6;9), t(9;11) or t(11;19), or mutations in receptor kinases, in key signaling mediators, proto-oncogenes, or epigenetic enzymes, e.g., mutations in *FLT3* (FMS-like tyrosine kinase 3), *TP53*, *c-KIT* or *IDH1/2*, *NPM1* (nucleophosmin 1), and CCAAT enhancer-binding protein (*CEBPA*) [1,2,123]. These mutations in AMLs have an impact on the choice of the most suitable therapy. 

The association between autophagy and recurrent genetic alterations has been described in several studies in AML, but needs further investigation [124,125]. Here, we summarize and update the recent advances that have highlighted the link between autophagy and fusion genes and recurrent oncogenic mutations in AML and the involvement of autophagy in chemotherapy treatment (Figure 2).

#### 4.4.1. Fusion Genes in AML and Autophagy

Most cases of APL are caused by a de novo t (15;17) (q22; q21) translocation, which results in the fusion of the *RARA* gene with the *PML* gene [126,127]. APL cells that have a lower expression of autophagy-related genes than normal cells have a reduced autophagic activity. By using differentiating agents, such as all-*trans* retinoid acid (ATRA) and arsenic trioxide (ATO) currently used in clinical settings, the expression level of autophagy-related genes increases, thus restoring autophagy in APL cells [128]. Both agents can activate ETosis, a type of cell death mediated by the release of neutrophil extracellular traps (ETs). In addition, mTOR-dependent autophagic action is required for ATO-induced NETosis in APL cells [129]. Of note, rapamycin, the inhibitor of mTOR pathway, synergizes with ATO in the eradication of leukemia-initiating cells (LIC) through the activation of NETosis in both APL cells and an in vivo APL model [129].

Mixed lineage leukemia (*MLL*) gene translocations 11q23 were observed in approximately 80% of pediatric AML. In these, the *MLL* gene can, by genomic translocation, be fused with >60 different fusion partners [130]. Treatment with the RAS oncogene inhibitor, tipifamib, leads to the inhibition of AML with the t(6;11) translocation by inducing both apoptosis and autophagy [131]. Another study demonstrated that ATG5 participates in the development of *MLL-AF9*-driven leukemia, but not in AML-sensitive chemotherapy mice expressing MLL-AF9 [132]. 

Acute myeloid leukemia with core binding factor (*CBF-AML*) is characterized by the presence of t(8;21) (q22; q22), or inv (16) (p13q22)/t(16;16), which leads to the formation of *RUNX1/RUNX1T1 (AML1/ETO)* and *CBFbeta-MYH11*, respectively [133]. The activation of ULK1-mediated autophagy may control and delay *AML1-ETO9a* -guided leukemogenesis in an AML *CASPASE-3* knockout mouse model [134], suggesting that CASPASE-3 is an important regulator of autophagy in AML. The results of these studies highlight the different roles of autophagy in the initiation, progression, and chemotherapeutic responses in AML cells, depending on the different type of aberrant oncoprotein.

#### 4.4.2. Genetic Mutations in AML and Autophagy


*
**FLT3**
*


Among the most common genetic alterations in AML, the tyrosine kinase 3 (*FLT3*) gene mutation occurs in approximately 30% of AML cases.

The most frequent aberrations affecting *FLT3* gene, associated with a poor prognosis in AML, are the internal tandem duplication (*FLT3*-ITD) in the juxtamembrane domain, and point mutations, involving the tyrosine kinase domain of *FLT3* (*FLT3*-TKD) [135]. *FLT3*-ITD expression increases basal autophagy in AML cells through a mechanism involving transcription factor ATF4 (activating transcription factor 4) [136]. In addition, the inhibition of autophagy in *FLT3*-TKD cells, which are resistant to the FLT3 inhibitor quizartinib (AC220), also inhibits proliferation both in vitro and in vivo [136]. More recently, the acquired D835Y mutation induced resistance to the FLT3 inhibitor sorafenib, and activated autophagy in *FLT3*-ITD-positive cell lines. By inhibiting autophagy, the authors were able to overcome resistance to sorafenib in *FLT3*-ITD-positive AML, improving its efficacy [137]. Recently, a study showed that the inhibition of autophagy reduces the repopulation potential of *FLT3*-ITD AML LSCs associated with mitochondrial accumulation [138]. In addition, the authors showed that autophagy inhibition improves p53 activity and increases the TKI-mediated inhibition of AML progenitors [138].

Autophagy not only contributes to downstream proliferation of the FLT3-ITD receptor, but may also be involved in mutated receptor degradation. In fact, in one study, the frequent activation of the receptor tyrosine kinase RET was observed in several AML subtypes [139]. RET mediates autophagy suppression in an mTORC1-dependent manner, leading to the stabilization of the mutant FLT3 receptor. The genetic or pharmacological inhibition of RET decreased the growth of FLT3-dependent AML cells, with the upregulation of autophagy and *FLT3* depletion [139]. These results suggest that restoring autophagy in FLT3-dependent AML may result in the degradation of mutated FLT3, and therefore may represent an interesting therapeutic approach. It has also been shown that the inhibition of the FLT3-ITD protein leads to an increase in ceramide synthesis and mediates ceramide-dependent mitophagy, leading to AML cell death [140,141]. 


**
*KIT*
**


*KIT* mutations are associated with an increased proliferation of leukemic cells and an increased risk of AML recurrence [142,143]. A recent study reported that the *KIT* D816V mutation in AML cells increases basal autophagy, stimulating AML cell proliferation and survival via STAT3 signaling [144]. A different point mutation in c-*KIT* (N822K T > A) constitutively activates this receptor, making AML cells highly sensitive to sunitinib (a tyrosine kinase inhibitor), resulting in AML cell death through the activation of both apoptosis and autophagy processes [145].


*
**NPM1**
*


Mutations in *NPM1* (nucleophosmin 1) are the most frequent genetic alterations in adult AML, responsible for the aberrant localization of the NPM1 protein in the cytoplasm [146]. Increased autophagic activity found in *NPM1* -mutated AML cells is involved in leukemic cell survival [147]. Mutant *NPM1* can also interact with the tumor suppressor protein PML (leukemia pro-myelocytic protein), leading to PML delocalization and stabilization that, in turn, can activate autophagy via AKT signaling [147]. In another study, it was shown that in AML patients carrying mutant *NPM1*, the glycolytic enzyme PKM2 (pyruvate kinase M2) induced autophagy via phosphorylation of the autophagic protein Beclin 1, contributing to cell survival [148]. Finally, the NPM1 mutant protein can also interact with the autophagic protein ULK1, stimulating the TRAF6-dependent ubiquitination of ULK1 via miR-146, thereby maintaining ULK1 stability and functionality and promoting autophagic cell survival [149]. Furthermore, it was observed that the expression of RASGRP3, a protein associated with tumor progression, is upregulated in patients with AML with *NPM1* mutation compared to patients with AML without mutant *NPM1*. The authors demonstrated that *NPM1* -mut blocks the degradation of the RASGRP3 protein through binding to the ubiquitin ligase E3 MID1 protein, leading to RASGRP3 overexpression, as well as promoting the downstream activation of EGFR-STAT3, which in turn promoted proliferation and autophagy in AML cells [150]. 


**
*P53*
**


Alterations in the tumor suppressor gene *TP53* are found in about 5–15% of AML cases, and, frequently, in older patients [151,152]. It has been proposed that the role of autophagy in the development of AML can be determined by the status of *TP53*. For wild-type *TP53* AML, researchers have shown that pharmacological autophagy blockade achieves therapeutic benefits, while AMLs harboring *TP53* mutations do not respond to the inhibition of autophagy by hydroxychloroquine (HCQ) [153,154]. The use of autophagic inhibitors may be a potential therapeutic strategy to use, particularly for the treatment of *TP53* wild-type AML. For AML with *TP53* mutations, autophagic pathways may be a therapeutic option to use for the elimination of mutant T *TP53*.

Another study demonstrated that macroautophagy stimulation by 17-AAG, a HSP90 inhibitor, causes the degradation of TP53 R248Q in AML cells and also enhances the transcription of autophagy-associated genes [155]. In addition, accumulated evidence indicates that TP53 activated by a variety of cellular stresses can trigger autophagy through the transactivation of pro-autophagy genes, including *DRAM1* (autophagic modulator regulated by DNA damage 1), *SESN1* (sestrin 1), and (sestrin *SESN2*) [155,156,157,158]. 

A recent study highlighted the role of autophagy in AML cells, in the context of p53-mediated apoptosis, which is associated with increased cytotoxicity to treatment with MDM2 inhibitors and Ara-C when miR-10a is inhibited [159]. The antileukemic strategy based on the use of MDM2/X inhibitors in wild-type p53 tumors to restore the normal and active conformation of p53, MDM2, and MDMX has not been extensively tested [160]. Thus, the use of a combination of treatments, including MDM2 inhibitors with autophagic modulators, may be a new strategy to improve the treatment of wild-type *p53* AML.

Pharmacological treatments that modulate autophagy in AML patients carrying *p53* mutations participate in the degradation of aberrant p53 proteins. The point mutation of *TP53* at the amino acid residue R428 (R248Q), with gain-of-function activity, gives rise to malignant activity in lung cancer cells [161] and a loss of tumor suppressor function in AML [162].

Interestingly, the treatment with the Hsp90 inhibitor (17-AAG) results in the activation of chaperone-mediated autophagy, which induces the degradation of the aberrant protein p53R248Q in AML cells. In particular, under conditions of metabolic stress, 17-AAG induces the interaction between p53R248Q and the chaperone protein Hsc70, triggering chaperone-mediated autophagy to degrade p53R248Q [155]. These data open new opportunities for future studies that may elucidate the functional involvement of different types of autophagy and their connection with molecular mechanisms to improve anticancer therapies against AML harboring the different *TP53* variants.


***IDH1/2* (isocitrate dehydrogenase)**


Recent advances in bioinformatics have enabled the identification of several epigenetic mutations affecting AML, including *IDH1/2*, Tet methylcytosine dioxygenase 2 (*TET2*), DNA methyltransferase 3A (*DNMT3A*), and *ASXL1*, all of which are associated with the pathogenesis of AML [163,164,165]. IDH proteins are isocitrate dehydrogenases, implicated in various biological processes, such as energy metabolism, histone demethylation, DNA modification, and adaptation to hypoxia. Further studies are needed to investigate innovative therapies based on targeted autophagy in combination with DNA hypomethylation to treat AMLs harboring certain types of epigenetic alterations.


*
**DNMT3A**
*


Mutations in the *DNMT3A* gene, an enzyme involved in the methylation of CpG dinucleotides, are present in 20–23% of adult patients with de novo AML [165]. Several studies have shown that the treatment of AML patients with DNA methyltransferase inhibiting agents, such as azacitidine (5-aza-2′-deoxycytidine), induces autophagic activity in AML leukemia cells [166]. A study performed on a *DNMT3A* R878H conditional knock-in mouse model, used to predict the specific long non-coding RNAs (lncRNAs) regulated by *DNMT3A* mutations in AML, first identified 23 differentially expressed *lncRNAs*, then the downstream target genes regulated by these lncRNAs, including *ATP6V1A*, a critical autophagy-related gene, the overexpression of which is associated with poor prognosis in AML [167]. However, there is still little evidence of a direct involvement of *DNMT3A* gene mutations with autophagic activity in AML.

Further studies are needed to understand the functional significance of autophagy associated with different genetic mutations in AML cells. 

## 5. Autophagy and Tumor Microenvironment (TME) in AML

HSCs primarily reside until their maturation in the bone marrow microenvironment (BMM), which is composed of multiple different cell types including osteoblasts, mesenchymal stem cells, adipocytes, endothelial cells, fibroblasts, macrophages, and hematopoietic cells, and provides several signals to regulate and support the production of billions of blood cells necessary to maintain homeostasis. The most primitive stem cells in the bone marrow are maintained in hypoxic niches that regulate the fate of these cells and hematopoietic progenitor cells in terms of quiescence, self-renewal, and differentiation [168]. Leukemic cells can infiltrate the niches leading to their enhanced self-renewal and proliferation, quiescence, and resistance to chemotherapeutic agents. The adaptation of leukemic cells to the BMM is important in the clonal selection that leads to leukemia progression, drug resistance, and relapse in AML. BMM has long been known to support LSCs survival and chemotherapy resistance [169]. 

A recent study has described an autophagy derived from the BM/tumor microenvironment (TME), called host autophagy [170]. Host autophagy and leukemic blast-related autophagy are important for both tumorigenesis and the regulation of tumor cell metabolism, a crucial component of therapy resistance [170]. Among several processes, autophagy participates in the regulation of TME, which is then involved in cancer progression [171]. In this context, an increased expression of ATG5 was observed in the mesenchymal stem cells of AML patients, and, by silencing *ATG5* in AML cells, the chemosensitivity to the genotoxic agents of these cells increased [172]. Taken together, these data highlight the significance of host autophagy activity in the development of AML.

## 6. Autophagy Affects AML Metabolism

In order to meet their high metabolic demands, proliferating leukemia cells undergo extensive reprogramming of cellular energy metabolism [86,173]. Metabolic reprogramming in tumor cells is an adaptive mechanism consisting in the ability to consume more glucose than non-proliferating cells, and results in increased lactate secretion, even in the presence of oxygen [174,175]. This mechanism allows for rapid adaptation in response to the high energy demands needed to sustain the high rate of proliferation in AML. Patients with a high level of gene expression of the glycolysis cycle and TCA (tricarboxylic acid cycle) showed a negative clinical outcome [176]. LSCs, in addition to glucose, can also use fatty acids and amino acids such as glutamine in order to provide precursors of the TCA cycle to support mitochondrial metabolism in LSCs [177].

Previous researchers have shown that normal HSCs, LSCs, and AML blasts have distinct and unique metabolic profiles. Leukemic blasts preferentially use metabolic pathways such as glycolysis [176] and the pentose phosphate pathway [178] to generate all the macromolecular elements needed to maintain the proliferative state [179,180].

LSCs, on the other hand, preferentially use mitochondrial metabolism to maintain their quiescence and self-renewal capacity. Notably, LSCs use mitochondrial oxidative phosphorylation (OXPHOS), not glycolysis, to generate ATP for survival [181,182].

A study indicated that chemotherapy-resistant LSCs, after chemotherapy induction, present a high level of phosphorylation; the inhibition of OXPHOS restores the Ara-C sensitivity of these cells, pointing to mitochondrial oxidative metabolism as a potential target to eradicate these chemo-resistant LSCs [183]. As previously reported, mitophagy plays a critical role in maintaining mitochondrial activity in LSCs [95,96]. Autophagy also contributes to leukemogenesis by ensuring fatty acid oxidation by lipophagy activation, thus promoting mitochondrial OXPHOS [184].

Importantly, LSCs can use both fatty acids and amino acids to supply macromolecules to the tricarboxylic acid (TCA) cycle and to support mitochondrial metabolism [185,186]. 

Recent reports have highlighted the presence of specific alterations of mitochondrial metabolism in AML cells. AMP-activated protein kinase (AMPK) is a critical metabolic checkpoint kinase that balances cellular energy state with proliferation by stimulating catabolism, including glucose uptake, glycolysis, fatty acid oxidation, and autophagy, and inhibiting anabolism. Most studies using small molecule AMPK activators have reported the antitumor activity of these compounds in various cancer models [187]. AMPK activation reduces mTORC1-induced protein biosynthesis and fatty acid biosynthesis through the repression of acetyl CoA carboxylase phosphorylation. In AML cells, mTORC1 is constitutively activated, and targeting this pathway could be a therapeutic opportunity. In this sense, a previous study had shown that the pharmacological inhibition of mTORC1, or preventing its activation through the removal of glutamine, induced the activation of autophagic flow [188]. Unexpectedly, the activation of autophagy is involved either in the death or in the survival of AML cells, according to the concentration of the inhibitor used. Indeed, a low concentration of the mTORC1 inhibitor, AZD8055, activates cytotoxic autophagy, while a high concentration of AZD8055 or the use of L-asparaginase, which harbors glutaminase activity and inhibits mTORC1, induces cytoprotective autophagy [188,189]. The metabolic alterations, such as increased glucose or amino acid uptake, impact mTOR signaling [190]. Then, the cross-talk between cancer metabolism and autophagy activation leads to opposite effects on cell survival, which are drug concentration-dependent, and these need further investigation to aid in the development of novel therapeutic strategies.

In AML, some mutations are also associated with intracellular metabolic changes, such as internal tandem duplication (ITD) mutations of *FLT3* associated with increased glycolytic activity through the upregulation of the gatekeeper glycolytic enzyme hexokinase 2 (HK-2) [191]. The heterozygous deletion of *ATG5* leads to increased proliferation in vitro and more aggressive leukemias in an *MLL-ENL* “in vivo” model of AML [103].

Then, the metabolism in AML cells mainly depends on OXPHOS to keep the cell proliferation rate and the metabolic demand high, while leukemic blasts use different metabolic pathways, including glycolytic metabolism.

The greater propensity for the use of glycolysis in AML, determined by adaptation, the greater the expression of monocarboxylate transporters (MCT) implicated in the transport of pyruvate, lactate, and ketone bodies. Among various members of the MCT family, MTC1 and MTC4 are of particular interest for their high expression found in various cancers, including AML, associated with a poor prognosis [192,193,194]. Our recent studies demonstrated that the inhibition of MCT1, MCT4, or their chaperone CD147 impairs leukemia cell proliferation, and pointed to CD147 and MCT4 as potential therapeutic targets in AML. The use of the inhibitors AC-73 (for CD147) and syrosingopine (for MCT4) in the treatment of AML induces autophagy, alone or in combination with conventional chemotherapeutic agents, cytarabine (Ara-C) and all-trans-retinoic acid (ATRA), resulting effective in overcoming the chemotherapy resistance of treated AML cells [193,194]. The tumor’s fructose 1,6-bisphosphatase 1 (FBP1), a gluconeogenesis regulatory enzyme, has been demonstrated to inhibit aerobic glycolysis in tumor cells. Increased expression levels of FBP1 in leukemic cell lines cause an induction of cell death and decrease the proliferation rate in FBP1-MV4-11 blasts. FBP1-overexpressing blasts may initiate leukemia cell death by activating mitochondrial reprogramming in AML cells, supporting the therapeutic contribution of FBP1-based drugs for AML treatment [195].

Thus, targeting autophagy in combination with drugs that block tumor metabolic reprogramming may be a winning strategy to overcome resistance to common chemotherapeutics in AML.

However, the role of autophagy in metabolic reprogramming responsible for therapy resistance in AML cells should be better delineated. For example, Ara-C-resistant AML cells show increased fatty acid β-oxidation (FAO) and oxidative phosphorylation OXPHOS, characterized by high mitochondrial ATP production and oxygen consumption rate (OCR) [183]. Consequently, blocking the metabolism of OXPHOS or FAO in AML cells can overcome resistance to Ara-C [185,196,197].

Tumor autophagy and host TME autophagy are both important for tumorigenesis and regulate cancer cell metabolism by influencing each other’s activity to support tumor growth [177]. Host autophagy, similar to tumor autophagy, could regulate TME metabolism as a mechanism of resistance to therapy. In fact, metabolic reprogramming most often occurs in the tumor microenvironment (TME) and participates in cancer growth and therapy resistance in numerous types of cancer, and thus is important [198]. In particular, bone marrow stromal cells (BMSCs) can reprogram their metabolism to support tumor growth and resistance, promoting AML cell progression and chemoresistance in vitro and in vivo, by increasing the activity of the OXPHOS and TCA cycle of leukemia cells or by providing aspartate to fuel pyrimidine synthesis [199]. The CXCL12/CXCR4 pathway may play a crucial role through mTOR activation [200]. The blockade of mTOR by rapamycin inhibited the stimulating effect of stromal cells on the glycolysis rate in AML blasts [200]. Overall, these data further underlie the importance of the metabolism in the TME in cancer growth and chemotherapy resistance. However, more studies are needed to elucidate the functional role of host autophagy and its association with therapy resistance and processes underlying therapies targeting both AML and TME cells.

## 7. Autophagy and Chemotherapy in AML

The pharmacological targeting of autophagy, combined with chemotherapy, appears to hold great promise in developing more effective treatment strategies for curing AML (Figure 3). In particular, the molecular mechanisms involved in autophagic therapies differ according to the type of disease, tumor stage, and type of therapy used, as they also interact with other cellular processes essential for cell survival, such as apoptosis. However, two opposite types of action are distinguished in autophagic therapies: those that induce the onset of autophagy to activate autophagic cell death (ACD) [201] (Table 1); and those that inhibit autophagy by reactivating chemosensitivity in drug-resistant cells (Table 2).

At present, chloroquine and hydroxychloroquine are the only autophagy inhibitors approved by the U.S. Food and Drug Administration for clinical use. They are being tested, in combination with conventional chemotherapeutics or targeted agents, in several completed or ongoing studies for the treatment of many types of tumors (http://clinicaltrials.gov/), including hematopoietic malignancies [202,203]. In this context, a phase II clinical trial combining hydroxychloroquine (HCQ) with imatinib in CML (NCT01227135) is noteworthy. However, other Phase I/II clinical trials indicate that the problems of specificity and potency with hydroxychloroquine will need to be addressed with more selective and potent autophagy inhibitors for optimal anticancer treatment.

Regarding the treatment of AML, the cytotoxic effect of autophagy inhibitors HCQ and/or baflomycin A1 (BafA1) is most evident in U937 cell lines Ara-C chemoresistant [204] and KG-1 compared to HL-60 cells when treated with ATO and ATRA [205]. Furthermore, the in vivo therapeutic effects of BafA1 with Ara-C support their potential as novel therapies for AML [206]. Furthermore, the combination of an autophagy inhibitor (chloroquine or LY294002) and a CDK4/6 inhibitor synergistically activated AML apoptosis [207].

Several inhibitors and small molecules have been studied to have autophagy-inhibiting activity. The VPS34-IN1 inhibitor, a specific inhibitor of the vacuolar protein VPS34 [208], was recently developed and its autophagy suppressive activity was tested in AML cells. Notably, VPS34-IN1 inhibits basal and L-asparaginase-induced autophagy in AML cells. Furthermore, VPS34-IN1 has also been shown to impair mTORC1 signaling in these cells [209]. Another promising strategy in targeting autophagy is the development of agents or small molecules that bind the LIR motifs of autophagic receptors with high affinity. In this context, agents that bind to the LIR motifs of three selective cargo receptors (OPTN, p62 and NDP52) have been shown to restore the sensitivity of AML cells to Ara-C [210].

Several molecules and drugs have been characterized for autophagy-activating properties. Dihydroartemisinin, a derivative of artemisinin used in the treatment of malaria, inhibits AML cell growth by triggering the ferroptosis of leukemic cells in an iron-dependent manner. Dihydroartemisinin-mediated autophagy acts by regulating AMPK/mTOR/p70S6k signaling and is critical in ferritin degradation, the accumulation of ROS, and ferroptotic cell death [211]. In addition to these inhibitors, we have shown that the antileukemic effect of a small molecule AC73 and Syrosingopine, alone or in combination with Ara-C or ATO, is mediated at least in part by the activation of autophagy in AML cells [194].

Other agents that induce autophagy in AML cells are currently under investigation in combination approaches, including statins, S100A8, and arginase [212,213,214]. Statins act on a crucial enzymatic step of cholesterol biosynthesis, decreasing the availability of cholesterol in cells and promoting the activation of autophagy [212]. To activate autophagy, S100A8 physically interacts with the autophagy regulator BECN1, whose expression is increased in AML-resistant cells. However, S100A8 depletion sensitizes leukemia cells to chemotherapy [213].

Arginine depletion, using a recombinant form of human arginase I (rhArg), inhibits the growth of AML blasts due to their dependence on extracellular arginase, but is also able to increase autophagy in these cells, justifying a strategy of combining it with 3-methyladenine or CQ to selectively target leukemic cells [214].

Multiple agents in development, in preclinical and clinical studies, and even many clinical drugs have been found to activate cytoprotective autophagy (Table 1), including mTOR inhibitors, kinase inhibitors, natural products, and antiangiogenic agents, resulting in a cytoprotective effect that increases chemoresistance [215,216]. Several studies have shown that autophagy can be activated after treatment with deoxycytidine analogues [88,206], BET inhibitors [217], mTOR catalytic inhibitors [87], histone methyltransferase inhibitors [218], and BCL2 inhibitors [109], actually contributing to the drug resistance of LSCs in AML. Another example has been demonstrated with the treatment of AML cells with Ara-C, anthracyclines, or sorafenib, which are able to activate and increase autophagic flow in these cells, including LSCs, allowing them to resist chemotherapy [16,155,159,160]. In this case, it may be useful to use autophagy inhibitors combined with chemotherapy to bypass this induced resistance.

**Table 1 cells-12-01553-t001:** Autophagy inducers in AML.

Name	Mechanism	References
Cytotoxic		
Dihydroartemisinin	mTOR/p7056k-inhibition/AMPK-activation	[207]
AC-73	CD147 inhibitor	[190]
Syrosingopine	MCT4 inhibitor	[189]
Statins	Cholesterol biosynthesis	[208]
S100A8-S100A9	BECN1	[209]
Arginase	Arginine-depleting enzyme	[210]
dendrogenin A (DDA)	Metabolite of cholesterol	[215]
Arsenic Trioxide (ATO)	Fusion protein degradation	[126]
All-trans retinoic acid (ATRA)	Fusion protein degradation	[126]
Cytoprotective		
Rapamycin	mTOR inhibitor	[127]
Sorafenib	Multi-targeted RTK	[134,136]
Azacitidine (AZA)	DNA methyltransferase	[135,217]
BIX-01294	Histone methyltransferase G9a	[214]
Cytosyne arabinoside (ARA-C)	DNA synthesis	[16]
Daunorubicin (DNR)	DNA synthesis	[216]
BET inhibitors	Decrease MYC, BCL2, and BCL6	[213]
Venetoclax	BCL2 inhibitors	[107]

The table reports some of the small molecules and/or agents capable of inducing a cytoprotective or cytotoxic autophagic response in the context of AML treatment.

**Table 2 cells-12-01553-t002:** Autophagy inhibitors in AML.

Name	Mechanism	Target Point	References
3-Methyladenine	PI3-K inhibitor	Autophagosome	[210]
LY294002	PI3-K/mTOR inhibitor	Autophagosome	[203]
SBI0206965	ULK1 inhibitor	Autophagosome	[216]
Bafilomycin A1	V-ATPase inhibitor	Autophagolysosome	[201,202]
VPS34-IN1	VSP34 inhibitor	Autophagolysosome	[204,205]
Roc325	Unknow	Lysosome	[217]
Lys05	Unknow	Lysosome	[136]
Cloroquine	Unknow	Lysosome	[203]
Hydroxychloroquine	Unknow	Lysosome	[198]
XRK3F2	SQSTM1/p62	Mitophagy	[15]

The table reports some autophagy inhibitors described in the review.

Recently, a metabolite of cholesterol in mammals, dendrogenin A (DDA), has been shown to have the same effect as Ara-C, the first-line chemotherapy used in clinics for AML. DDA induces cell death through the inhibition of a sterol isomerase, thereby disrupting cholesterol metabolism and activating lethal autophagy. This metabolite also strengthens the antineoplastic activity of anthracyclines (idarubicin or daunorubicin, DNR), which represent, in combination with Ara-C, one of the pillars of AML therapy [219]. However, in these cases, targeting autophagy by genetic or pharmacological means, combined with chemotherapy, appears to hold great promise in developing more effective treatment strategies.

As seen with Ara-C, anthracycline DNR is also described to trigger cytoprotective autophagy via the ULK1-AMPK pathway and independently of mTORC1. Thus, a selective ULK1 kinase inhibitor (SBI0206965) improved DNR cytotoxicity in vitro [220]. Another drug frequently used in AML patients is the hypomethylating agent azacitidine (Aza). Recently, Aza treatment has been shown to induce a cytoprotective effect of autophagy in AML cells. The novel lysosomal autophagy inhibitor ROC-325, a derivative of chloroquine, potentiates the antileukemic activity of Aza in in vitro and in vivo studies, indicating the use of autophagy inhibitors combined with chemotherapy to bypass the Aza-induced resistance [166]. Furthermore, as autophagy participates in the resistance to tyrosine kinase inhibitor (TKI) treatment, the combination of TKIs with the autophagy inhibitor Lys05 could improve *FLT3*-ITD AML treatment [138].

## 8. Conclusions

The numerous studies conducted in recent years strongly support the view that targeting autophagy, and altered metabolism, represents novel opportunities to further explore for the treatment of leukemias, including AML. However, it remains essential to investigate further how leukemic and host autophagy promote or prevent the resistance to chemotherapies, as their molecular mechanisms still remain poorly understood. Furthermore, most of the studies performed so far point to the use of drug combinations as a rational design in the future of antileukemic treatment. Thus, the identification of predictive autophagic and metabolism biomarkers should help to identify those stratified AML patients who may benefit from a novel therapy, taking into account AML subtype, oncogenic drivers, and the clonal heterogeneity of AML.

## Figures and Tables

**Figure 1 cells-12-01553-f001:**
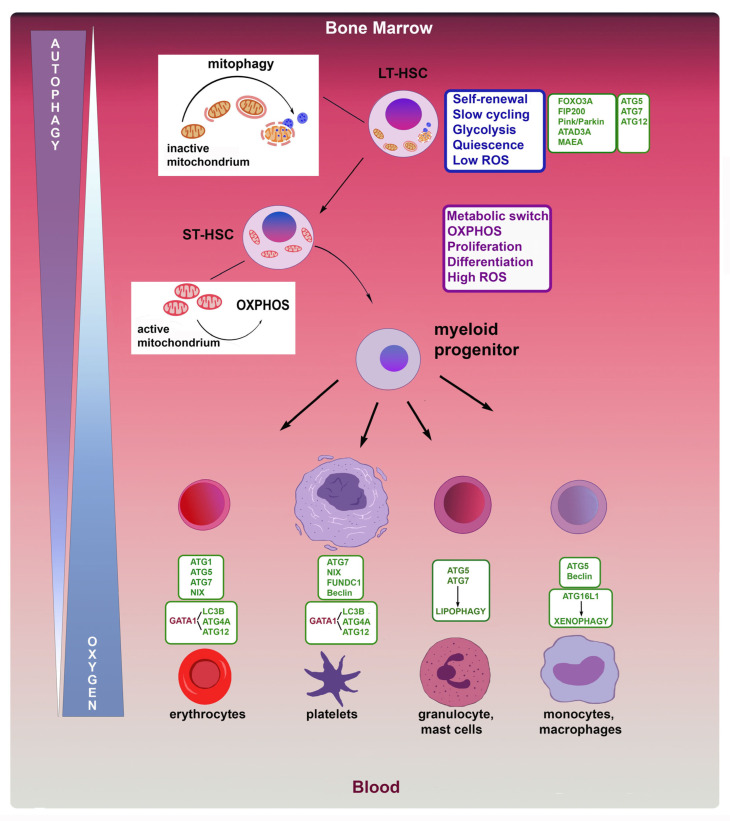
Autophagy and metabolism during normal hematopoiesis. Autophagy-mediated regulation of metabolism is required for self-renewal, differentiation, and lineage-specific functions of hematopoietic stem cells. Long-term HSCs utilize anaerobic glycolysis and suppression of mitochondrial oxidative phosphorylation by mitophagy to maintain quiescence and stemness under hypoxic conditions in the bone marrow. The metabolic transition to oxidative phosphorylation results in a switch from long-term HSC to short-term HSC, cell cycle entry, and differentiation. The components of the autophagic machinery required for effective differentiation, maturation and/or function of terminal cells of the different lineages are indicated in green.

**Figure 2 cells-12-01553-f002:**
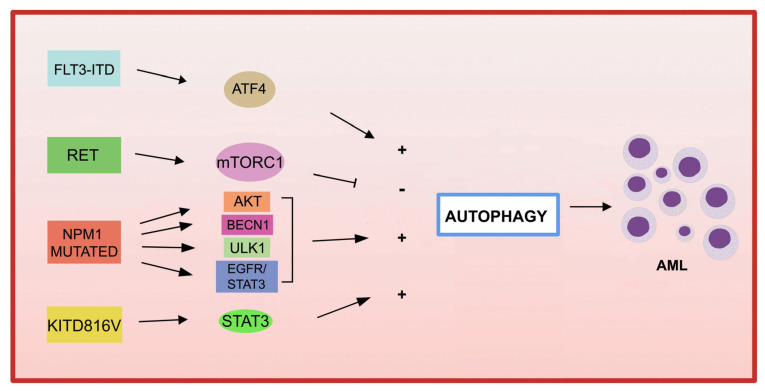
Genetic mutations in AML and autophagy. Several recurrent genetic abnormalities in AML are involved in autophagy deregulation, leading to leukemia progression.

**Figure 3 cells-12-01553-f003:**
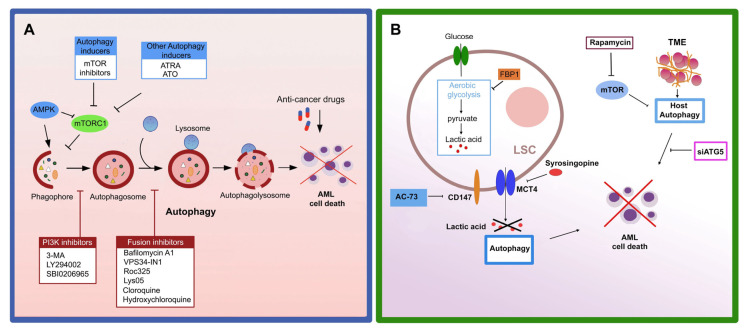
Autophagy modulation in targeted therapy for AML. (**A**) Autophagy inducers or inhibitors to use in combination with common chemotherapy drugs for the treatment of AML are indicated in left panel. (**B**) New molecules and drugs that regulate autophagy, through the modulation of AML metabolism and tumor microenvironment, are indicated in right panel. Abbreviations: ATO, arsenic trioxide; ATRA, all-trans retinoic acid; 3-MA, 3Methyladenine; siATG5, siRNAs targeting ATG5; TME, tumor microenvironment.

## Data Availability

Data sharing not applicable.

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
