# Peer review of "Advances in Understanding the Links between Metabolism and Autophagy in Acute Myeloid Leukemia: From Biology to Therapeutic Targeting"

_cells, 2023, doi:10.3390/cells12111553_

Round 1

Reviewer 1 Report

The manuscript by Saulle and colleagues explores the interesting question of the therapeutic relevance of autophagy in acute myeloid leukemia (AML) treatment. The manuscript is formatted as a review and provides a general overview of AML, followed by an introduction into autophagy, roles of autophagy in hematopoietic stem cells (HSC) and differentiation into the various myeloid cell populations. The authors then discuss the potential roles of autophagy in AML, including leukemic stem cells, but also in relation to several of the most commonly observed mutations, impact on the tumor microenvironment, AML metabolism and the interplay between autophagy and chemotherapy. These are all important aspects of AML, including relapsed disease. The manuscript is generally informative and should be of interest to hematologists, however there are some points that should be addressed to improve the manuscript.

1. The authors should provide a reference for the statement on lines 194-195 : "HSCs, although the are mostly inactive, contain a higher number and major volume...."

2. Figure 1 is useful, and although it is clear that the review is focused on AML, I still think it would be useful to at least mention the (potential) roles of autophagy in the lymphoid cells as well.

3. In the genetic mutation in AML section, DNMT3A mutations are commonly detected in AML and should be included here. 

4. Addition of one or two figures to illustrate some key points, such as showing mechanisms of how certain mutations common to AML may support the idea of targeting autophagy pathways to treat AML. Similarly, a figure could incorporate how different drugs regulate autophagy and how this could be useful to treat AML, modulate AML metabolism and modify the tumor microenvironment.

1. The use of English needs to be improved in several instances. Here are just a few examples:

line 66:   "where cargoes is degraded.."

line 72: "individual unfold proteins"

line 149: "neighbour", but the rest of the manuscript is in American English

line 190: "associated to a metabolic shift"

Line 210: "LKB1 also participates to..." 

Line 217: "In mice depleted of FIP200 resulted in increased..."

Line 303: "by means of through pharmacological..."

Line 327: "In ATG5-knockout mouse"

I would suggest to make two sentences out of the text on lines 334-338. For example, the first sentence could end at "autophagy." and the second sentence begin with "Pretreating".

Line 375: "Then several...."

Line 441: "associated to poor"

Line 645: "a host spot mutation"

Line 751: "in several cancer"

Line 783: "these data further underlies"

Reviewer 2 Report

In this review, the authors review the role of autophagy and the impact of its deregulation on the metabolism of normal and leukemic hematopoietic cells. The authors focus on autophagy-dependent regulation of AML development and relapse and point out the increasing therapeutic potential of interventions in autophagy.

The review has a logical structure and is focused on role of autophagy in HSC/ malignant blood cells and mentions the relevant literature in this field. I have only some minor points, which should be addressed in more detail, to help the reader to put the points being made by the authors in better focus.

Minor remarks

1)   Page 3: line 96-160: The authors nicely present the process of macroautophagy in detail. An additional illustration would help to better understand the complex interplay of ULK1 /mTOR signaling.

2)   Pages 7 and 8: Saulle et al. elucidate autophagy’s role in different cell types of the blood. Here, the authors should explain shortly in every single paragraph, the function of the cell types mentioned.

Round 2

Reviewer 1 Report

The authors have mostly addressed all points raised earlier. It is curious that the new figures are in the supplemental section of the manuscript. What is the reason to not put them in the main section of the manuscript?

In table 1, all-trans retinoic acid is spelled incorrectly (at least retinoid is not commonly used for ATRA).

Author Response

Dear Reviewer,

In the 2nd revised version of our manuscript, we have loaded Legend Figures, Figure 2 and Figure 3 at the end of the manuscript in the 2nd revised version of the review.  

As suggested by the reviewer, we have written “all-trans retinoic acid” in Table 1.